# A High-Throughput NMR Method for Lipoprotein-X Quantification

**DOI:** 10.3390/molecules29030564

**Published:** 2024-01-23

**Authors:** Erwin Garcia, Irina Shalaurova, Steven P. Matyus, Lita A. Freeman, Edward B. Neufeld, Maureen L. Sampson, Rafael Zubirán, Anna Wolska, Alan T. Remaley, James D. Otvos, Margery A. Connelly

**Affiliations:** 1Labcorp, Morrisville, NC 27560, USA; emgarcia20ub@gmail.com (E.G.); shalaui@labcorp.com (I.S.); matyuss@labcorp.com (S.P.M.); 2Translational Vascular Medicine Branch, National Heart, Lung, and Blood Institute, National Institutes of Health, Bethesda, MD 20892, USA; litaf@mail.nih.gov (L.A.F.); neufelde@nhlbi.nih.gov (E.B.N.); rafael.zubiran@nih.gov (R.Z.); anna.wolska@nih.gov (A.W.); alan.remaley@nih.gov (A.T.R.); jimotvos@gmail.com (J.D.O.); 3Clinical Center, National Institutes of Health, Bethesda, MD 20892, USA; msampson@cc.nih.gov

**Keywords:** lipoprotein-X, LCAT deficiency, cholestasis, hypercholesterolemia

## Abstract

Lipoprotein X (LP-X) is an abnormal cholesterol-rich lipoprotein particle that accumulates in patients with cholestatic liver disease and familial lecithin–cholesterol acyltransferase deficiency (FLD). Because there are no high-throughput diagnostic tests for its detection, a proton nuclear magnetic resonance (NMR) spectroscopy-based method was developed for use on a clinical NMR analyzer commonly used for the quantification of lipoproteins and other cardiovascular biomarkers. The LP-X assay was linear from 89 to 1615 mg/dL (cholesterol units) and had a functional sensitivity of 44 mg/dL. The intra-assay coefficient of variation (CV) varied between 1.8 and 11.8%, depending on the value of LP-X, whereas the inter-assay CV varied between 1.5 and 15.4%. The assay showed no interference with bilirubin levels up to 317 mg/dL and was also unaffected by hemolysis for hemoglobin values up to 216 mg/dL. Samples were stable when stored for up to 6 days at 4 °C but were not stable when frozen. In a large general population cohort (*n* = 277,000), LP-X was detected in only 50 subjects. The majority of LP-X positive cases had liver disease (64%), and in seven cases, had genetic FLD (14%). In summary, we describe a new NMR-based assay for LP-X, which can be readily implemented for routine clinical laboratory testing.

## 1. Introduction

Lipoprotein-X (LP-X) is an abnormal lipoprotein particle enriched in phospholipids (60–65 mol%) and non-esterified cholesterol (22–25 mol%) but low in the content of neutral lipids (triglycerides and cholesteryl esters) [1,2,3]. Unlike the micellar structure of spherical lipoproteins, LP-X has a multilamellar vesicular structure with a variable diameter of 30–70 nm [4,5,6,7], which does not contain apolipoprotein B (apoB) as is normally found on low-density lipoprotein (LDL) and very low-density lipoprotein (VLDL) particles [3]. It is also relatively devoid of exchangeable apolipoproteins of the type found on high-density lipoprotein (HDL) [3]. Owing to the absence of apoB, LP-X is not cleared through the LDL receptor but rather by phagocytic uptake by macrophages and mesangial cells in the glomerulus of the kidney [8].

LP-X is observed in patients with familial lecithin–cholesterol acyltransferase (LCAT) deficiency (FLD), a rare autosomal recessive disorder caused by mutations in the LCAT gene [9,10,11,12]. LCAT is a key enzyme in the esterification of free cholesterol and, as such, plays a central role in the maturation of HDL particles and in reverse cholesterol transport [13,14,15]. Patients with FLD present with low HDL cholesterol and anemia and can develop end-stage kidney disease, most likely from the deposition of LP-X in the kidney [16,17].

LP-X is commonly found in patients with cholestasis resulting from a wide variety of liver diseases but most often in patients with primary biliary cholangitis (formerly referred to as primary biliary cirrhosis or PBC) [1,18,19,20], primary sclerosing cholangitis (PSC) [20,21], graft versus host disease [22], and occasionally during pregnancy [23]. LP-X likely forms because of the secondary deficiency of LCAT in cholestatic liver disease and because of the reflux of biliary cholesterol into the plasma compartment [21]. In addition, LP-X may also be transiently formed in patients shortly after intralipid infusion [24,25].

Despite the large number of disorders and conditions associated with LP-X and its potential diagnostic importance, there is currently no clinical laboratory method amenable to high-throughput clinical testing. LP-X can be the major carrier of plasma cholesterol, and unlike LDL cholesterol, levels of LP-X cholesterol are not reduced by statin treatment. Patients with elevated levels of LP-X can develop severe cutaneous xanthomas, and the only effective treatment is its physical removal by plasmapheresis [26,27,28]. The first methods for detecting LP-X clinically involved labor-intensive lipoprotein precipitation and centrifugation methods [29,30,31,32,33]. LP-X can also be identified by agarose gel electrophoresis based on its unusual cathodal migration compared to other lipoproteins [1]; however, lipoprotein gel electrophoresis is no longer widely used by clinical laboratories. In addition, Sudan black, the neutral lipid stain most often used for lipoprotein electrophoresis, poorly stains LP-X. Recently, more sensitive stains, such as filipin, a fluorescent polyene antibiotic that specifically binds free cholesterol [34], and BODIPY-cholesterol [35], have been described for improving the detection of LP-X by gel electrophoresis. LP-X can also be indirectly estimated by the ratio of total cholesterol/apoB [36].

A variety of methods based on ^31^P [37], ^13^C [38], and ^1^H [39] nuclear magnetic resonance (NMR) spectroscopy have been described for detecting LP-X, but these involve significant sample preparation and/or are performed on NMR instrumentation not routinely used by clinical laboratories. The current report describes a new method to detect and quantify LP-X via a modified version of the *NMR LipoProfile^®^* test analysis used for routine clinical measurement of VLDL, LDL, and HDL particles and their different-size subclasses [40]. The measurement is conducted using the Vantera^®^ Clinical Analyzer (Labcorp, Burlington, NC, USA), a 400 MHz NMR instrument designed specifically for high-throughput clinical laboratory analysis of lipoproteins and small molecule metabolites [40].

## 2. Results

### 2.1. Quantification of LP-X

The region from 0.72 to 1.02 ppm in the proton NMR spectrum of serum or plasma, which includes the lipoprotein particle lipid methyl group signals, is used to quantify LP-X. Figure 1A shows a representative fit of the broad lipid methyl signal envelope for a normal plasma sample analyzed using the standard lipoprotein deconvolution model. As is typical when there are no unusual lipoproteins present in the sample, there is very close correspondence (virtual superposition) between the measured signal (orange) and the signal calculated from the sum of the derived contributions made by the constituent lipoprotein subclasses (black). Figure 1B is a representative example of the poor fit obtained for a sample containing LP-X when the standard lipoprotein deconvolution model is used. In such cases, there is a clear mismatch between the measured and calculated signals that results in a substantial residual signal (blue). When the measured residual in this region of the spectrum exceeds a prespecified value, the software flags the presence of LP-X and prompts a reanalysis using a modified deconvolution model that includes a reference standard signal for LP-X.

Figure 2A shows the results of using the modified LP-X deconvolution model to analyze the same sample from Figure 1B that could not be successfully analyzed using the standard lipoprotein model. The measured (orange) and calculated (black) signals are now virtually superimposed, and the residual signal is negligible. The derived amplitude of the LP-X component signal, the shape and position of which are shown in Figure 2B, provides the LP-X concentration. To enable the reporting of LP-X concentrations in mg/dL cholesterol units, a calibration curve was used (Figure 2C) that was produced by spiking a serum sample with varying amounts of synthetic LP-X containing a fixed amount of cholesterol as described in the Methods section. Synthetic LP-X, which was designed to match the lipid composition of endogenous LP-X, is a multilamellar vesicle that, like endogenous LP-X, has cathodal migration on agarose gel electrophoresis [34,35] and, when administered to LCAT-deficient mice, induces kidney injury [41].

### 2.2. Assay Performance

The intra-assay imprecision was assessed by testing three pools of varying LP-X levels using five replicates in a single run. The inter-assay imprecision involved testing the same pools in duplicate for 2 runs per day over 5 days. The imprecision results are summarized in Table 1. The %CV for the low, medium, and high pools range from 11.3 to 15.4%, 2.9 to 3.6%, and 1.5 to 1.8%, respectively.

The assay was evaluated for its ability to accurately detect and quantify LP-X. The average value obtained when testing blank samples was 0 mg/dL cholesterol units (LOB). The functional sensitivity or lower limit of quantitation (LLOQ), determined by replicate testing of 7 pools over 3 days, was 44 mg/dL. To evaluate linearity, nine pools were used containing LP-X, which covered the range observed in clinical samples. Regression analysis was performed on the measured and expected values. Linearity was demonstrated for the following range: 89–1615 mg/dL cholesterol units.

### 2.3. Stability and Substance Interference Testing

The stability of LP-X, as measured by NMR in serum, was evaluated for up to 6 days at 4 °C. Measurements were deemed acceptable if they were within 10% of the baseline values. Results demonstrated that LP-X is stable for 6 days when stored at 4 °C (Table 2); however, freezing results in an apparent decrease in LP-X.

Two endogenous substances that are commonly found to interfere with test results in clinical samples, bilirubin and hemoglobin, were tested for possible interference with the LP-X assay. Each substance was tested at multiple concentrations (conjugated bilirubin up to 317 mg/dL and hemoglobin up to 216 mg/dL) to determine at what level the substance might elicit a >10% change in the LP-X assay results. The results indicated that neither substance caused significant interference with test results.

### 2.4. Simultaneous Quantification of LP-X and LP-Z

A plasma sample from a patient with acute alcoholic hepatitis, a disease in which both LP-X and LP-Z have been shown to be present, was tested for LP-X and another abnormal lipoprotein particle called LP-Z. LP-Z is a triglyceride and a free-cholesterol enriched, small-size LDL particle that has recently been shown to be a prognostic marker in patients with liver failure due to acute alcoholic hepatitis [42]. As shown in Figure 3, the signal from the residual plots for LP-X and LP-Z do not significantly overlap, thus allowing their separate and simultaneous quantification when both are present in the same sample.

### 2.5. LP-X Disease Associations

The frequency of LP-X was examined by reanalyzing digitally stored NMR spectra that were collected from April 2018 to August 2021 from the NMR testing of non-frozen serum samples from patients for which the *NMR LipoProfile* was ordered for routine testing in the Labcorp^®^ clinical laboratory. The assay detected LP-X in 1024 of the 1,080,663 (0.09%) spectra analyzed. The distribution of LP-X in these samples is shown in Table 3.

The association of LP-X with disease was examined in a second large general cohort of patients (*n* = 277,000) by determining the primary and secondary diagnoses of patients for whom LP-X was detected. A total of 77 samples were found to contain LP-X. After eliminating repeat specimens, 50 unique patients were identified containing LP-X (Table 4). Of this population, 52% were female with a mean age of 45.5 years, 24% were White, 14% were Black/African American, 2% were Asian, and the remainder were unknown. The main identifiable cause of LP-X was liver disease (64%), with cirrhosis, drug-induced, and graft versus host disease being the most common diagnoses. Seven cases (14%) had familial LCAT deficiency or fish-eye disease, which also has low LCAT activity from genetic mutations [43]. For the remainder of the cases, no clear cause was identified, or the presence of LP-X was associated with a disease not previously described as having the presence of LP-X.

## 3. Discussion

The ability of NMR spectroscopy to simultaneously measure resonances arising from components of a mixture enables the quantification of the analytes in a complex bio-sample without physically separating them. NMR has been used to quantify the lipoprotein particles in serum and plasma samples on a clinical NMR analyzer [40]. Here, we describe a new NMR-based assay to detect and quantify abnormal lipoprotein LP-X, using a non-negative least squares deconvolution algorithm. The software assesses the agreement of fit between the methyl lipid resonance of the sample and the composite of the methyl lipid resonances of the lipoprotein components in the standard lipoprotein deconvolution model by detecting the residual signal. When a poor fit (disagreement or mismatch) in the data is observed, as would be the case when LP-X is present in the sample owing to its unique spectroscopic fingerprint, it is flagged for the presence of LP-X. The software assay subsequently prompts the use of a modified model to quantify the abnormal lipoprotein.

The developed assay described here appears to have adequate sensitivity and demonstrates linearity over biologically relevant concentrations of LP-X. No interference was observed with the addition of bilirubin and hemoglobin; however, other endogenous and exogenous (over-the-counter drugs) substances were not tested for possible interference with the LP-X assay. In terms of stability, samples with LP-X may be stored at 4 °C for up to 6 days. Unfortunately, LP-X does not seem to be stable with freezing and thawing, as a significant bias was observed in the LP-X results even after one freeze/thaw cycle. This, however, is consistent with what other investigators have previously reported [34]. It is hypothesized that the structure of LP-X is disrupted when ice crystals form in its aqueous core during freezing [34].

To demonstrate the robustness of the developed assay in detecting LP-X, even in the presence of another abnormal lipoprotein LP-Z as occurs in acute alcoholic liver disease [44] and obstructive jaundice [2], a sample obtained from a patient with high bilirubin was analyzed. Both LP-X and LP-Z components were detected and quantified by the deconvolution algorithm, and the residual was minimal, proving that the software effectively detects and quantifies both abnormal lipoproteins.

The LP-X assay was able to identify the presence of LP-X in the general population. As expected, LP-X was primarily found in patients with cholestatic liver disease from a wide variety of causes and was also observed in patients with genetic LCAT deficiency. We also detected its presence in a small number of patients with other types of diagnoses not previously reported to have LP-X. In some cases, this may be due to a false positive test result and will have to be more carefully evaluated in a larger study in the future.

The newly developed assay for LP-X has several potential practical applications. In fact, the algorithm for detecting LP-X is now routinely used in the NMR analysis by Labcorp to improve the quantification of lipoprotein particle numbers. When LP-X is present, an alternative algorithm is used to deconvolute the NMR spectra to obtain more accurate lipoprotein particle concentrations and lipid values. Because the presence of high LP-X has been reported to cause interference in other routine laboratory tests, the new test could have other applications. Pseudohyponatremia [45,46], as well as pseudohypokalemia and pseudohypochloremia [46] have been described in patients with cholestasis. LP-X was also reported to interfere with LDL cholesterol results, typically leading to false elevations [47,48,49]. It has also been described that LP-X caused interference in ApoE phenotype analysis [20]. The LP-X assay described here may, therefore, be used as an effective tool in the clinical laboratory for detecting LP-X in these types of patients to reduce laboratory errors and avoid incorrect clinical management (e.g., addressing electrolyte abnormalities). The presence of LP-X could also be added to the laboratory report. This may be particularly useful for helping to identify FLD patients, who are frequently missed and not diagnosed until they develop end stage renal disease [50]. Another potential application is to monitor progress with the treatment of recombinant LCAT therapy in patients with FLD. A recombinant LCAT enzyme (ACP-501) was shown to be safe and well-tolerated in a Phase I clinical trial [51]. Although the effect of recombinant LCAT on LP-X was not examined in this study, LP-X has been used ex vivo to monitor recombinant LCAT treatment [34].

Several other practical advantages of the NMR-based LP-X assay deserve further mention. First, the sample preparation is minimal, involving only mixing of the sample with phosphate buffer, performed onboard by the analyzer (automated), as opposed to performing tedious fractionation or lipoprotein precipitation, use of expensive reagents, or performing signal quenching experiments. Second, the sample-to-results turnaround is fast (1.5 min), which is amenable to high-throughput screening applications. Third, the assay described here has a sensitivity and linearity that is not only adequate for clinical testing but potentially also for monitoring LP-X levels in response to recombinant LCAT therapy. Samples for testing, however, are limited to non-frozen specimens due to the interference caused by freezing consistent with previous observations [34].

A potentially important limitation of the NMR assay is that the position of the lipid methyl NMR signal for LP-X is largely dependent on its chemical composition and, likely, lipid configuration and particle size, whereas separation of LP-X by agarose gel electrophoresis depends on both electro-endosmosis and particle size, with detection dependent on staining methodology [34]. Given the physical and chemical heterogeneity of LP-X and the unknown correspondence between the two methods at this time, any samples identified as being positive for LP-X by NMR should be confirmed by agarose gel electrophoresis or perhaps another method, if important in the clinical management of a patient. With that said, in a single case report of drug-induced liver injury leading to cholestasis, NMR-measured LP-X levels appeared to correlate with the clinical presentation, suggesting that the NMR test can be used as a simple means to detect LP-X particles in a patient sample and enable a more sophisticated interpretation of unusually high cholesterol levels. However, future work is necessary to assess the correspondence between the NMR assay and other assays that detect LP-X and to establish the clinical utility of this assay [52].

## 4. Materials and Methods

### 4.1. Synthetic LP-X and Natural LP-X from High Bilirubin Samples

The formation of synthetic LP-X (NIH, Bethesda, MD, USA) was previously described [41]. Briefly, 24 mole % cholesterol (Avanti Polar Lipids, Inc., Alabaster, AL, USA) was combined with L-α-lecithin (Avanti Polar Lipids, Inc., Alabaster, AL, USA) in chloroform and dried under nitrogen. The dried mixture was then resuspended in phosphate-buffered saline and sonicated to generate multilamellar particles.

NMR spectra from samples containing natural LP-X were obtained from discarded, non-frozen, de-identified clinical serum samples with total bilirubin ≥8 mg/dL. After collecting the ^1^H NMR spectrum, a sample was flagged for LP-X by applying the deconvolution algorithm described below.

### 4.2. Explanation of Deconvolution Models

The *NMR LipoProfile* spectral deconvolution approach relies on the assumption that “the whole (plasma or serum NMR signal) is the sum of its (lipoprotein signal) parts”. Specifically, the measured plasma/serum lipid signal envelope “whole” centered at 0.8 ppm is computationally decomposed into the spectral amplitude “parts” contributed by the component VLDL, LDL, and HDL subspecies, thereby providing their particle concentrations. The standard lipoprotein deconvolution model, which accounts for the contributions of all normal lipoprotein subspecies, analyzes >99% of patient samples successfully as judged by achieving a “match” (high degree of correspondence) between the shape and amplitude of the measured plasma/serum signal and the one calculated from the sum of the derived lipoprotein components. Rare samples for which a “mismatch” is observed between measured and calculated plasma signals are those that contain abnormal lipoproteins that make spectral contributions not accounted for by the standard lipoprotein deconvolution model. LP-X is one such abnormal lipoprotein, and its presence in patient samples can be flagged by observing such a deconvolution mismatch. LP-X in samples identified in this way can be readily quantified by employing a modified deconvolution model that includes a reference standard for LP-X.

### 4.3. NMR Data Acquisition

The assay that quantifies LP-X utilizes a ^1^H NMR spectrum collected for a plasma or serum sample tested on a Vantera^®^ Clinical Analyzer (Labcorp, Burlington, NC, USA) [40]; the spectrum is acquired identically to that obtained for the *NMR LipoProfile* test. Each specimen is diluted (1:1 (*v*/*v*)) with NMR diluent onboard and injected into the detection cell. The Vantera Clinical Analyzer is equipped with a 400 MHz (9.4 T) Agilent spectrometer, a 4 mm indirect detection probe, and a fixed flow cell that was equilibrated at 47 °C via a variable temperature control module (Agilent Technologies, Santa Clara, CA, USA). Details for this spectral acquisition and analysis (~1.5 min total time) have been described previously [40].

### 4.4. LP-X Quantification by Deconvolution

The Fourier-transformed spectrum is the input to the quantification algorithm. It is initially fed into a deconvolution analysis involving a library of methyl signal lineshapes corresponding to normal VLDL, LDL, HDL, and their subclasses, as well as the protein background components (standard lipoprotein model). The lineshape deconvolution was achieved using a non-negative least squares fitting program [53]. For a sample with normal lipoprotein particles, the methyl signal envelope coincides closely with the composite signal of the lipoprotein and protein components in the library, resulting in minimal to no residual signal. However, in the presence of abnormal lipoproteins, which elicit different spectroscopic fingerprints, a large residual is observed due to the envelope features not accounted for by the available lipoprotein/protein components in the algorithm. We utilized the residual to flag the presence of abnormal lipoproteins when it exceeds a predefined threshold, and to prompt the use of a modified deconvolution model to quantify LP-X. Because the accumulation of LP-X may be accompanied by the appearance of another abnormal lipoprotein species called LP-Z, as has been observed in patients with acute alcoholic liver disease [44] and obstructive jaundice [2], we incorporated the methyl lipid signal of LP-Z as an additional component in the modified model. Although not much is known about LP-Z, it has been described as a triglyceride and non-esterified cholesterol-enriched particle with diameters in the small LDL range (18.5–21.5 nm) [2]. Adding a reference standard signal for LP-Z in the modified model enabled the software to detect its presence in a sample. Furthermore, the LP-Z chemical shift (centered at 0.81 ± 0.01 ppm) is well-resolved from that of LP-X (centered at 0.84 ± 0.01 ppm). This allowed the algorithm to effectively quantify both particles simultaneously when present in a plasma/serum sample.

The methyl lipid signal of the LP-X component (including the cholesterol and phospholipids present in LP-X [39]) was generated from synthetically prepared particles as described above, while the LP-Z component was isolated by ultracentrifugation and gel filtration from a cholestatic patient with liver disease awaiting liver transplant. The resulting LP-X amplitude from the deconvolution algorithm was transformed into cholesterol concentration units (mg/dL) by using an empirically determined conversion factor. The conversion factor was obtained by calculating the LP-X amplitude from spectra collected for pooled serum spiked with different amounts of synthetic LP-X and relating the peak signal areas by NMR to the chemically-measured total cholesterol on a Roche Diagnostics Cobas C701 Analyzer (Roche, Basel, Switzerland). The description and analytical characteristics of the LP-Z assay are presented in a separate publication [42].

### 4.5. Assay Analytical Performance: Imprecision, Sensitivity, and Linearity

Serum pools (*n* = 3) targeting low, intermediate, and high natural LP-X concentrations were used to determine assay imprecision. Within-lab imprecision consisted of duplicate tests run twice per day for 5 days (*n* = 20) on one Vantera instrument, while within-run imprecision was calculated from five replicates for the same pools collected in a single run. Mean and %CV were calculated for each pool.

Serum pools were used to determine the limit-of-blank (LOB) and lower limit-of-quantitation (LLOQ). For the LOB, five serum pools were used, which were analyzed in 4 replicates collected over 3 days. For LLOQ, seven pools with varying low levels of natural LP-X were prepared and were analyzed in four replicates over 3 days. Mean concentration and coefficients of variation (CVs) were calculated for each pool. The LLOQ was determined at the LP-X concentration corresponding to 20% CV.

A series of mixtures of low, medium, and high serum samples containing natural LP-X were created to evaluate assay linearity. The expected values came from measuring the concentrations of the low, medium, and high pools using the Vantera Clinical Analyzer and calculating the intermediate concentrations based on the combinations that were made between these samples. Each mixture was analyzed twice. Deming regression analysis was performed on the average of the measured versus the expected LP-X values.

### 4.6. Stability (Refrigerated, Freeze–Thaw)

Refrigerated stability was assessed by testing serum pools containing low, intermediate, and high natural LP-X at 0 (baseline), 1, 4, 5, and 6 days in duplicate. For the effect of the number of freeze–thaw cycles, eight samples with different natural LP-X concentrations were subject to 2X freeze–thaw cycles before testing in duplicate. Stability is claimed at the time point or # of cycles prior to the LP-X results exhibiting at least two consecutive time points with bias exceeding 10% relative to the baseline value.

### 4.7. In Vitro Test for Interfering Substances

Two of the most common endogenous substances, conjugated bilirubin and hemoglobin, were tested for possible assay interference. Stock solutions (20X) were prepared in H_2_O. During the initial screening, half of the serum pool was spiked with the stock solutions at the following dose of the potential interferent: 8.9 mg/dL conjugated bilirubin (Calbiochem/EMD Millipore Corp., Temecula, CA, USA), 42 mg/dL hemoglobin (Sigma-Aldrich, St. Louis, MO, USA. The other half was spiked with H_2_O to serve as control. The pool contains 247 mg/dL LP-X prepared by spiking with synthetic LP-X. Natural LP-X from high bilirubin samples was not suitable for this testing due to the presence of naturally occurring bilirubin. Analysis was performed using three replicates for each pool. Interference was determined at the substance concentration exhibiting >10% bias on LP-X results.

### 4.8. LP-X Disease Associations

The association of LP-X with various diseases was determined in a large general cohort (*n* = 277,000) of patients for which an NMR lipoprotein profile was requested for routine diagnostic testing at the NIH from June 2015 to July 2021. The de-identified database of spectra was collected under a clinical protocol (#93-CC-0094) for assay development in the Department of Laboratory Medicine, NIH.

## 5. Conclusions

Herein is described an automated and high-throughput method to detect and quantify LP-X by NMR spectroscopy that involves minimal sample preparation. The analytical performance of the assay shows that it can reliably measure circulating LP-X in clinical samples. Not only can this assay be used as an alternative method to detect LP-X, but it also has the potential to be used as a screening and monitoring tool in clinical and research settings.

## 6. Patents

Patent Number: US11467171B2, publication date: 2022-10-11.

## Figures and Tables

**Figure 1 molecules-29-00564-f001:**
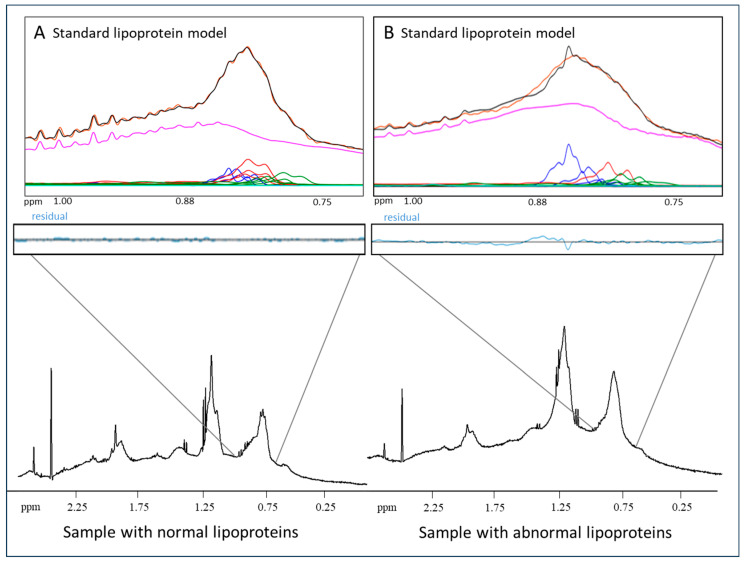
Quantification of LP-X using a standard NMR model: (**A**) the NMR spectrum of a normolipidemic plasma sample without LP-X, and (**B**) the NMR spectrum of a sample with relatively high LP-X. The composite of the methyl lipid signals of the lipoprotein components is illustrated with black lines (calculated) and orange lines (measured). The background signal, including the combined signal from the proteins, is in purple, and the signals corresponding to the VLDL, LDL, and HDL subclasses are in blue, red, and green, respectively. The residuals are illustrated in blue below the spectrum.

**Figure 2 molecules-29-00564-f002:**
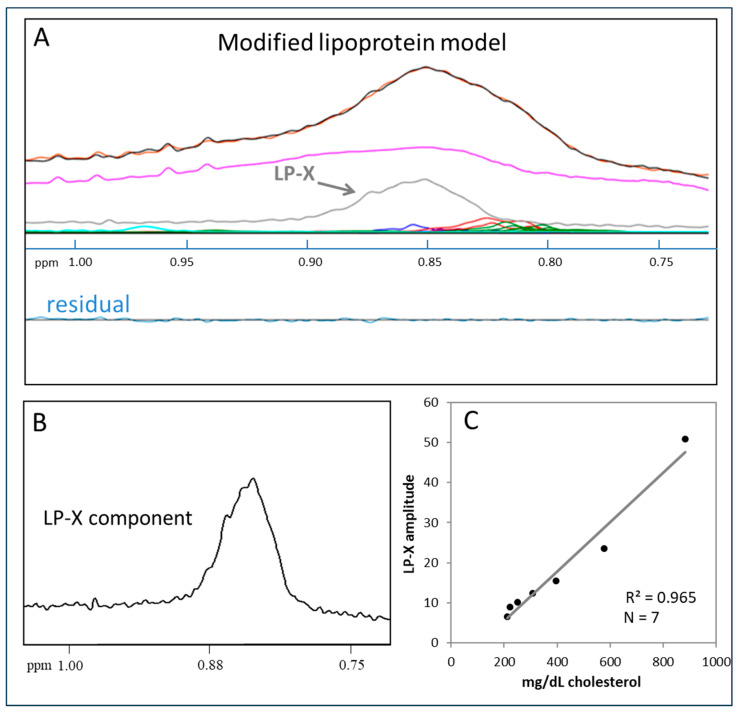
Quantification of LP-X with the use of a modified NMR model: (**A**) the NMR spectrum of a sample containing LP-X. The composite of the methyl lipid signals of the lipoprotein components (calculated) is illustrated with black lines and orange lines (measured). The LP-X component used to properly deconvolve the broad methyl lipid signal is illustrated in gray. The background signal, including the signals from the proteins, are in purple, and the signals corresponding to the VLDL, LDL, and HDL subclasses are in blue, red, and green, respectively. (**B**) Methyl lipid resonance of LP-X. (**C**) The relationship between the amplitude of LP-X by NMR of the cholesterol content of LP-X used to spike the sample.

**Figure 3 molecules-29-00564-f003:**
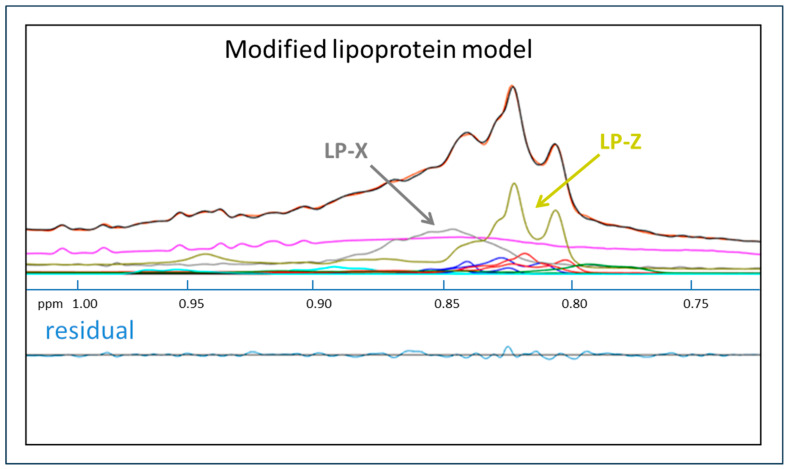
Quantification of LP-X and LP-Z with the use of a modified NMR model. The NMR spectrum from a sample collected from a patient with alcoholic hepatitis. The composite of the methyl lipid signals of the lipoprotein components (calculated) is illustrated with black lines and orange lines (measured). Signals from LP-X (gray) and LP-Z (chartreuse) are indicated. The background signal, including the combined signal from the proteins, is in purple, and the signals corresponding to the VLDL, LDL, and HDL subclasses are in blue, red, and green, respectively.

**Table 1 molecules-29-00564-t001:** Intra-assay and inter-assay imprecision for the NMR LP-X assay.

	LP-X (mg/dL Cholesterol)		
		Intra-Assay ^a^	Inter-Assay ^b^
Low pool	Mean	70.0	76.8
	SD	10.9	8.7
	%CV	15.4	11.3
Intermediate pool	Mean	329.1	328.1
	SD	9.6	11.8
	%CV	2.9	3.6
High pool	Mean	673.9	659.4
	SD	10.4	11.9
	%CV	1.5	1.8

^a^ Based on one run of five replicates. ^b^ Based on 2 runs per day in duplicate for 5 days (*n* = 20).

**Table 2 molecules-29-00564-t002:** The stability of LP-X at refrigerated temperature and multiple freeze–thaws.

Refrigerated				Freeze–Thaw			
Sample	Days	Mean	% Bias	Sample	Number of Freeze–Thaw Cycles	Value	% Bias
L1	Baseline	76.6	--	S1	Baseline	716.2	--
	1	74.8	−2.3		1X	442.8	−38
	4	80.8	5.6		2X	361.4	−50
	5	79.2	3.4	S2	Baseline	179.3	--
	6	77.6	1.3		1X	138.2	−23
L2	Baseline	328.7	--		2X	134.9	−25
	1	334.2	1.7	S3	Baseline	145.2	--
	4	320.7	−2.4		1X	79.1	−46
	5	336.1	2.3		2X	50.8	−65
	6	316.0	−3.9	S4	Baseline	256.6	--
L3	Baseline	667.6	--		1X	222.3	−13
	1	661.6	−0.9		2X	208.4	−19
	4	650.4	−2.6	S5	Baseline	62.3	--
	5	660.7	−1.0		1X	55.7	−11
	6	664.1	−0.5		2X	66.4	7
				S6	Baseline	348.1	--
					1X	271.1	−22
					2X	247.7	−29
				S7	Baseline	185.3	--
					1X	177.3	4
					2X	154.1	−17
				S8	Baseline	376.3	--
					1X	351.9	−7
					2X	317.6	−16

**Table 3 molecules-29-00564-t003:** Distribution of LP-X detected by NMR (*n* = 1024).

Percentile	LP-X (mg/dL Cholesterol)
100	2738
99	1301
95	239
90	149
75	94
50	53
25	16
10	4.9
2.5	2.2
0.5	0.4
0	0.1
Mean	95.5
SD	238

**Table 4 molecules-29-00564-t004:** Disease associations of LP-X in a large population cohort.

Specific Causes	Subjects (*n* = 50)	LP-X Level (IQR)
Drug-induced hepatic failure	5 (10%)	138 mg/dL (57–212)
Graft vs. host disease	5 (10%)	194 mg/dL (133–687)
Hepatitis C	3 (6%)	55 mg/dL (1.5–175)
Cirrhosis	16 (32%)	81 mg/dL (59–133)
Autoimmune hepatitis	3 (6%)	225 mg/dL (29–1023)
Unknown	5 (10%)	70 mg/dL (26–131)
Pancreatic disease	1 (2%)	8.5 mg/dL
Familial LCAT deficiency/fish-eye disease	7 (14%)	155 mg/dL (65–265)
Smith–Lemli–Opitz syndrome	1 (2%)	73 mg/dL
Referred as lipoprotein deficiency	4 (8%)	130 mg/dL (101–157)

## Data Availability

Data is available upon request.

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
