# Peer review of "A High-Throughput NMR Method for Lipoprotein-X Quantification"

_molecules, 2024, doi:10.3390/molecules29030564_

Round 1

Reviewer 1 Report

Comments and Suggestions for Authors

The authors describe how the LipoProfile NMR-based lipoprotein quantification method can be enhanced to detect and quantify the uncommon LP-X lipoprotein. This is of diagnostic significance as LP-X lipoprotein is not easily quantifiable through other methods. This is an interesting addition to the field of NMR-based lipoprotein analysis and for the most part the authors have set up and describe their research well. I only have a few nitpicks.

- The algorithm flags a spectrum when the residual exceeds a certain threshold. The authors should mention whether the presence of chylomicrons would also flag a spectrum.

- Table 4 should include the average LP-X concentrations for the individual causes.

- Check if the sample temperature of 47 degrees C is correct. Ref 40 mentions 45 degrees C.

- I do not understand the method of obtaining natural LP-X (section 4.1). This seems to imply that natural LP-X was separated from the serum, but this makes no sense in the context.

- Please confirm that the cohorts used in section 2.5 were from non-frozen samples, in particular the second cohort.

Author Response

The authors describe how the LipoProfile NMR-based lipoprotein quantification method can be enhanced to detect and quantify the uncommon LP-X lipoprotein. This is of diagnostic significance as LP-X lipoprotein is not easily quantifiable through other methods. This is an interesting addition to the field of NMR-based lipoprotein analysis and for the most part the authors have set up and describe their research well. I only have a few nitpicks.

- The algorithm flags a spectrum when the residual exceeds a certain threshold. The authors should mention whether the presence of chylomicrons would also flag a spectrum.

As a normal lipoprotein species that is present in post-prandial samples, chylomicrons are quantified using the LP4 algorithm.  The signals for chylomicrons and LP-X have very different shapes, therefore it is easy for the software algorithm to distinguish them from each other. The flag is for abnormal lipoprotein species, therefore, there is no flag for chylomicrons.

- Table 4 should include the average LP-X concentrations for the individual causes.

Good point.  We added the mean LP-X concentrations and interquartile ranges (IQR) to Table 4.

- Check if the sample temperature of 47 degrees C is correct. Ref 40 mentions 45 degrees C.

We have double-checked reference 40 and did not see a mention of 45°C. However, the flow cell is kept at 47°C in order to heat the sample to around 45°C at the time the NMR measurement is taken. So both temperatures are technically correct.

- I do not understand the method of obtaining natural LP-X (section 4.1). This seems to imply that natural LP-X was separated from the serum, but this makes no sense in the context.

We agree that the original sentence was confusing.  The sentence was reworded to state the following: “NMR spectra from samples containing natural LP-X were obtained from discarded, non-frozen de-identified clinical serum samples with total bilirubin ≥8 mg/dL.” (see lines 295-296)

- Please confirm that the cohorts used in section 2.5 were from non-frozen samples, in particular the second cohort.

Thank you for catching this.  The sentence was reworded to include the fact that the samples were non-frozen. “The frequency of LP-X was examined by reanalyzing digitally stored NMR spectra that were collected from April 2018 to August 2021 from NMR testing of non-frozen se-rum samples from patients for which the NMR LipoProfile was ordered for routine testing in the Labcorp® clinical laboratory.” (see lines 185-188)

Reviewer 2 Report

Comments and Suggestions for Authors

The present paper described design of a high-throughput NMR method to detect lipoprotein X (LP-X).  The presented NMR-based assay for LP-X was investigated on the point of linearity and sensitivity.

The following points should be considered for publication:

(1)    How long does it take to measure one sample using the presented high-throughput NMR method?

(2)     In Fig.1, the labels are “A, B”, but in captions, “a, b” are used (same as Fig. 2), which should be corrected.

 (3)    In Fig. 1 and Fig. 2, it is hard to distinguish the colored signals.  It is relatively easy to recognize blue and green spectra, but not easy for the others.  The orange spectra can be seen for this reviewer, and the red spectra cannot be recognized.  The way of presentation should be improved.

 (4)    In Fig. 3, it is easy to recognize spectra of LP-X and LP-Z with arrows, but it is hard to distinguish the other spectra.

Author Response

The present paper described design of a high-throughput NMR method to detect lipoprotein X (LP-X).  The presented NMR-based assay for LP-X was investigated on the point of linearity and sensitivity.

The following points should be considered for publication:

(1)    How long does it take to measure one sample using the presented high-throughput NMR method?

Acquisition of the NMR spectrum and the computational analysis using the LP4 algorithm (with or without the additional computation for the abnormal lipoproteins) for each sample is approximately 1.5 minutes. This information was added to section 4.3. (see lines 323-324)

(2)     In Fig.1, the labels are “A, B”, but in captions, “a, b” are used (same as Fig. 2), which should be corrected.

Thank you for catching this mistake.  The figure legends have been corrected.

 (3)    In Fig. 1 and Fig. 2, it is hard to distinguish the colored signals.  It is relatively easy to recognize blue and green spectra, but not easy for the others.  The orange spectra can be seen for this reviewer, and the red spectra cannot be recognized.  The way of presentation should be improved.

Unfortunately, there are only so many colors for the lines and we’ve found that this is the best way for us to illustrate all of the different spectra and the components.

 (4)    In Fig. 3, it is easy to recognize spectra of LP-X and LP-Z with arrows, but it is hard to distinguish the other spectra.

Please see the response to comment #3.  The most important take home message from Fig. 3 is that the signals for LP-X and LP-Z are different from each other as well as from the signals from normal lipoproteins which makes the signals easy to quantify using the software LP4 algorithm.